# Assessment of Antibody-Titer Changes after Second and Third Severe Acute Respiratory Syndrome Coronavirus 2 mRNA Vaccination in Japanese Post-Kidney-Transplant Patients

**DOI:** 10.3390/vaccines11010134

**Published:** 2023-01-06

**Authors:** Kumiko Fujieda, Akihito Tanaka, Ryosuke Kikuchi, Nami Takai, Shoji Saito, Yoshinari Yasuda, Takashi Fujita, Masashi Kato, Kazuhiro Furuhashi, Shoichi Maruyama

**Affiliations:** 1Department of Nephrology, Nagoya University Hospital, Nagoya 466-8560, Aichi, Japan; 2Department of Medical Technique, Nagoya University Hospital, Nagoya 466-8560, Aichi, Japan; 3Division of Clinical Laboratory, Gifu University Hospital, Gifu 501-1194, Gifu, Japan; 4Department of Nursing, Nagoya University Hospital, Nagoya 466-8560, Aichi, Japan; 5Department of Urology, Nagoya University Graduate School of Medicine, Nagoya 466-8560, Aichi, Japan; 6Department of Nephrology, Nagoya University Graduate School of Medicine, Nagoya 466-8560, Aichi, Japan

**Keywords:** SARS-CoV-2, vaccination, kidney transplantation, immunocompromised host

## Abstract

Post-renal-transplant patients have a relatively low antibody-acquisition rate following severe acute respiratory syndrome coronavirus 2 (SARS-CoV-2) mRNA vaccination. In this study, antibody titers were measured 5–6 months and 3 weeks to 3 months after the second and third SARS-CoV-2 mRNA vaccinations, respectively. Post-renal-transplant patients visiting our hospital who had received three SARS-CoV-2 mRNA vaccine doses were included in the study. SARS-CoV-2 immunoglobulin G antibody titers were measured three times: between 3 weeks and 3 months after the second vaccination, 5–6 months after the second vaccination, and between 3 weeks and 3 months after the third vaccination. A total of 62 (40 men and 22 women) were included, 44 of whom (71.0%) were antibody positive after their third vaccination. On comparing the antibody-acquired and antibody-non-acquired groups, body mass index (BMI, odds ratio [OR]: 1.44, 95% confidence interval [CI]: 1.07–1.93, *p* < 0.05) and the estimated glomerular filtration rate (eGFR, OR: 1.14, 95% CI: 1.06–1.24, *p* < 0.01) were associated with antibody acquisition. Therefore, in Japanese post-kidney-transplant patients, increases in the antibody-acquisition rate and absolute antibody titer after the third vaccination were observed, with BMI and eGFR associated with the antibody-acquisition rate.

## 1. Introduction

The coronavirus disease 2019 (COVID-19) pandemic, declared in early 2020, has resulted in significant mortality [1]. The mRNA vaccines that were developed for severe acute respiratory syndrome coronavirus 2 (SARS-CoV-2) have demonstrated high antibody-acquisition rates in healthy individuals, suggesting their usefulness in preventing infections and severe diseases [2,3,4,5]. However, immunosuppressed patients, particularly those who have undergone kidney transplantation, are more likely to develop severe SARS-CoV-2 infection than healthy individuals; however, the rate of antibody acquisition after SARS-CoV-2 mRNA vaccination is lower in this population [6,7]. Conversely, concerns regarding the sustainability of the vaccine persist. Furthermore, the effectiveness of vaccine protection reduces with time; therefore, boosters are recommended, especially for immunocompromised individuals [8,9,10]. In fact, several previous studies have revealed that additional vaccinations for post-kidney-transplant patients increase antibody titers, as compared with two vaccinations alone [11,12,13,14], and the effectiveness of additional vaccinations for post-transplant patients is becoming clear.

However, the kidney transplant situation in Japan differs from that in other countries. Many living-donor kidney transplants have been performed in Japan to overcome the issue of donor shortage, and ABO blood type (ABO)-incompatible kidney transplants have also been widely performed [15]. Furthermore, due to donor shortage, kidney transplants are performed in certain cases when weakly positive, donor-specific human leukocyte antigen antibodies are considered acceptable [15]. Therefore, blood purification, such as plasmapheresis, and stronger immunosuppression than that required in ABO-compatible transplants are necessary for ABO-incompatible kidney transplants. In addition, immunosuppression for renal transplantation may be stronger in Japanese patients than in those from other countries; therefore, obtaining Japan-specific data is necessary.

In our previous study, we measured anti-S SARS-CoV-2 immunoglobulin G (S-IgG) titers between 3 weeks and 3 months after the first two doses of the SARS-CoV-2 mRNA vaccine in post-kidney-transplant patients in Japan [16]. The results revealed that 31.5% of post-kidney-transplant patients acquired antibodies, which was not only significantly lower than that of healthy participants but also lower than the antibody-acquisition rates reported in post-kidney transplant patients overseas [17,18]. In this study, S-IgG titers were measured 5–6 months after the second vaccination and 3 weeks to 3 months after the third vaccination in post-kidney-transplant patients in Japan, and changes in the persistence of antibody titers and antibody-acquisition rate following the third vaccination were examined. Because S-IgG alone cannot dictate the ability to protect against infection, S-IgG and neutralizing antibodies (NT-IgG) do not necessarily correlate in the general population [19]; NT-IgG was also measured.

## 2. Materials and Methods

### 2.1. Patients

This study included post-kidney-transplant patients who visited Nagoya University Hospital between April and August 2022. In addition, patients who had received renal transplants at other hospitals and subsequently visited our hospital were also included. Of these, patients who had received three SARS-CoV-2 mRNA vaccine doses were eligible for inclusion. However, patients who had not been vaccinated due to anaphylaxis or a history of allergy were excluded. Furthermore, patients already on dialysis, those with post-COVID-19 infection, and non-consenting patients were also excluded. Data on patient background, past medical history, comorbidities, medications, and laboratory testing were collected. The Institutional Review Board of Nagoya University Hospital approved this study (approval number: 2010–1135 and 2020–0486), and all participants provided written informed consent. All methods were conducted in compliance with the principles of the Declaration of Helsinki and relevant guidelines.

### 2.2. Measurement of SARS-CoV-2 Antibody Titers

We measured three different SARS-CoV-2 antibodies for different purposes. First, we measured S-IgG, an antibody against the spike protein, which is elevated by both SARS-CoV-2 infection and vaccination. Patients with episodes of infection were excluded from the study to determine the effect of the vaccine. Additionally, we measured anti-N SARS-CoV-2 immunoglobulin G (N-IgG), an antibody against nucleocapsid, in the remaining cryopreserved specimens from patients with elevated S-IgG without clinical infectious episode to determine SARS-CoV-2 infection. Furthermore, since S-IgG alone is insufficient to determine antibody function, NT-IgG, which is a neutralizing antibody, was measured if any remaining specimens were available. Antibodies titers, such as N-IgG and NT-IgG, were assayed by LSI Medience Corporation.

Antibody titers were measured using sera obtained from blood tests on the day of the regular visit. The first measurement was taken 3 weeks to 3 months after the second vaccination, the second measurement was taken 5–6 months after the second vaccination, and the third measurement was taken 3 weeks to 3 months after the third vaccination. The analytical instrument used was a completely automated immunoassay analyzer (Alinity I; Abbott Laboratories, Chicago, IL, USA). A SARS-CoV-2 IgG II Quant Reagent Kit (Lot; 28531FN00, Abbott Japan Co., Ltd., Tokyo, Japan) was used for the antibody assay. The cutoff value was set at 50 AU mL^−1^ according to previous reports [20,21].

N-IgG antibody titers were measured on sera obtained from blood tests 5–6 months after the second vaccination if patients had elevated S-IgG despite no episodes of infection using an automated immunoassay analyzer (Architect i2000 SR; Abbott Japan Co., Ltd., Tokyo, Japan). An antibody, Architect SARS-CoV-2 IgG (Lot; 44367FN00, Abbott Japan Co., Ltd.), was used for the antibody assay. The cutoff value was set at 1.4 index according to the manufacturer’s protocol.

NT-IgG antibody titers were measured on sera obtained from blood tests 3 weeks to 3 months after the third vaccination using an automated immunoassay analyzer (iFlash 3000; Medical & Biological Laboratories Co., Ltd., Nagoya, Japan). An antibody, iFlash-2019-nCoV Nab (Lot; 20220501, Medical & Biological Laboratories Co., Ltd.), was used for the antibody assay. This assay is a one-step competitive immunoassay using direct chemiluminescence immunoassay. There is no ethical issue because the neutralizing assay was not performed using human-derived components, such as cell line, other than the sample. The cutoff value was set at 10.00 AU mL^−1^, according to the manufacturer’s protocol. Antibody titers with a result of 800 AU ml^−1^ or higher were treated as 800.

### 2.3. Statistics and Reproducibility

Baseline characteristics were presented descriptively and analyzed using the Mann–Whitney U and Fisher’s exact tests, depending on the data type. The distribution of antibody titers after 2 doses, 6 months after 2 doses, and after 3 doses is plotted in a violin plot. Univariate and multivariate logistic regression analyses were used to evaluate the factors associated with antibody acquisition. Odds ratios were shown in forest plots graphically. Model 1 was not adjusted. Model 2 was adjusted by age and sex during vaccination. Model 3 was adjusted by estimated glomerular filtration rate (eGFR) in addition to Model 2 for body mass index (BMI) and by eGFR in addition to Model 2 for BMI. For BMI and eGFR, single and multiple regression analyses were performed. Model 1 was single regression analysis. Model 2 included age and gender for multiple regression analyses. Model 3 was model 2 plus BMI for eGFR and eGFR for BMI. The correlation between S-IgG and NT-IgG is shown in the scatterplot. Correlation coefficients were also obtained. Statistical significance was set at *p* < 0.05. R software (R Foundation for Statistical Computing, Vienna, Austria, http://www.R-project.org/) accessed on 26 October 2022 was used for all statistical analyses in this study.

## 3. Results

### 3.1. Patient Background and S-IgG-Titer Trends

We investigated 62 transplant recipients (40 men and 22 women) who had received SARS-CoV-2 mRNA vaccines. The baseline characteristics of the patients in this study are presented in Table 1. The median age during vaccination was 54 years (interquartile range (IQR), 49–67 years), and that during transplantation was 50 years (IQR, 38–58 years). The median time from transplantation to vaccination was 84 months (IQR, 34–154 months). The median body mass index (BMI) was 22.9 kg m^−2^ (IQR, 21.4–26.2 kg m^−2^). Chronic glomerulonephritis was the most common primary renal disease (32.3%), followed by diabetic nephropathy (14.5%) and hereditary disease (12.9%). Diabetes mellitus and hypertension were comorbidities in 24.2% and 62.9% of the patients, respectively. The percentage of living-donor kidney transplants was high (90.3%), while ABO-incompatible kidney transplants accounted for 29.0%. Preemptive renal transplantation accounted for 37.1% (23 patients), and renal replacement therapy (RRT) before transplantation included hemodialysis (HD) and peritoneal dialysis (PD) in 33 and 6 patients, respectively. The median pre-transplant RRT period for HD and PD patients was 17 months (IQR, 7–73 months). Most patients used immunosuppressive drugs, predominantly prednisolone, tacrolimus, and mycophenolate mofetil. A total of 25 (40.3%) patients received all three BNT162b2 (Pfizer-BioNTech) doses, 12 (19.4%) received all three mRNA-1273 (Moderna) doses, 19 (30.6%) received a combination of BNT162b2 (Pfizer-BioNTech) and mRNA-1273 (Moderna), and 6 (9.7%) were unsure of the type of vaccine they received. Figure 1 shows the evolution of S-IgG titers in post-kidney-transplant patients. Among the post-kidney-transplant patients, 44.8% and 50.0% were found to have acquired S-IgG 3 weeks to 3 months and 5–6 months after the second vaccination, respectively. The S-IgG titer changed from positive to negative in one patient and from negative to positive in four at 5–6 months after the second vaccination. However, none of those four patients experienced episodes suggestive of SARS-CoV-2 infection. Considering the possibility of COVID-19 infection, we attempted to measure N-IgG for these four patients. However, only three patients were tested due to the unavailability of stored serum from one patient. All three patients were negative for N-IgG Appendix A. Seventy-one percent were found to have acquired S-IgG after the third vaccination, exhibiting a significant increase. Of the patients, 22.4% tested negative after the second vaccination but became positive after the third vaccination.

### 3.2. Comparison of the S-IgG-Acquired and -Nonacquired Groups

Clinical factors were compared between the S-IgG-acquired and -nonacquired groups at the time from 3 weeks to 3 months after the third vaccination. Table 2 shows the patient characteristics of each group. Significant differences in weight, BMI, eGFR, and primary disease were observed in each group.

### 3.3. Comparison of the Acquired S-IgG Titers among Vaccine Type

The S-IgG titers after the third dose of each vaccine type are shown in Figure 2. No significant differences were observed in S-IgG acquisition by vaccine type or combination.

### 3.4. Examination of Factors Associated with S-IgG Acquisition

Logistic regression analysis was performed to determine the factors involved in S-IgG acquisition. The analysis was performed to ensure that multicollinearity was not a challenge, and incorporated factors considered clinically meaningful are presented in Table 2. The results revealed that BMI and eGFR were significantly associated with S-IgG-acquisition rates, even after adjustment for various factors (Figure 3). However, for the primary disease, which was significantly different from S-IgG acquisition in the univariate analysis, no significant difference was observed in S-IgG acquisition in the multivariate analysis.

### 3.5. Regression Analysis for an Increase in S-IgG Titer from 5–6 Months after 2 Doses of Vaccination to 3 Doses of Vaccination

We showed that BMI and eGFR were associated with the rate of antibody acquisition. Subsequently, we investigated the correlation of these factors with the increase of S-IgG titers from 5–6 months after the second immunization to after the third immunization. Measurements of 58 patients are presented in Table 3. BMI showed no significant association in both single and multiple regression analysis, suggesting a less linear association. Similarly, eGFR showed no significant association but appeared to have a more linear association than BMI.

### 3.6. Comparison of NT-IgG and S-IgG after the Third Vaccination

Next, NT-IgG was measured to evaluate the ability of vaccination to protect against infection (Figure 4). NT-IgG and S-IgG were correlated in Japanese transplant patients, and most patients who had acquired S-IgG titers were also positive for NT-IgG titers.

## 4. Discussion

The S-IgG-acquisition rate among Japanese post-kidney-transplant patients 3 weeks to 3 months after the second vaccination was as low as 41.0%; nonetheless, the antibody-acquisition rate increased to 50.0% 5–6 months after the second vaccination. The proportion of patients who were found to have acquired antibodies after the third vaccination increased significantly to 71.0%, and their BMI and eGFR values were found to be associated with the rate of antibody acquisition after the third vaccination. Furthermore, NT-IgG and S-IgG were correlated. In healthy participants, sufficient antibody titers are acquired after two doses of vaccinations. However, they decline over time, and antibody titers are higher after three vaccinations than immediately after two vaccinations [22,23]. In kidney transplant recipients, antibody titers were insufficient after two vaccinations. However, they did not decline over time, and after three vaccinations, the highest antibody titers were obtained to that point. The trend was similar to that of healthy participants in that the highest antibody titer was obtained after the third vaccination. In kidney-transplanted patients, the results of this study revealed that the antibody-acquisition rate after the second vaccination (44.8%) was slightly higher than that reported in our previous study (31.5%) [16] and similar to that reported in other countries [17,18]. This may be because older adults, who are at a higher risk of severe SARS-CoV-2 infection, were prioritized for vaccination in Japan; therefore, the patients in the previous study were from an older age group than those in this study. In fact, the median age at vaccination in this study was 54 years, while that in the previous study was 61 years. Despite the differences in age categories and the timing of post-vaccination measurement of antibody titers in various studies [24,25,26,27], they all suggest that acquiring antibodies with increasing age is challenging. Although we did not find any difference in age between patients with and without antibody acquisition, and we cannot make direct comparisons with previous literature, we speculate that age may have a bias in the rate of antibody acquisition. It is also possible that age is statistically underpowered because the number of patients in this study was not large.

Next, considering the change in S-IgG titer approximately 6 months after the second vaccination, these results suggest that although the rate of antibody positivity due to vaccination is low in post-kidney-transplant patients, once positive, the antibody titer tends to be maintained. Only one patient who had previously acquired antibodies turned negative. In contrast, four patients acquired antibodies 5–6 months after the second vaccination. N-IgG titers were also measured in the three individuals for whom residual sera were available to verify the possibility that they were infected as asymptomatic carriers. However, the results were negative, and the possibility of infection was considered low. It is possible that patients who were not infected but whose exposure produced a booster effect and whose antibody titers were slightly below the threshold tested positive. Notwithstanding, no rapid attenuation of antibody titers was observed among the transplant patients.

Regarding S-IgG titers after the third vaccination, an increase in S-IgG titers was observed even in Japanese post-kidney-transplant patients, indicating that the effect of the additional vaccination was substantial. The S-IgG-acquisition rate after the third vaccination was similar to that reported overseas [12,13,14]. On comparing the group that acquired S-IgG after the third vaccination with the group that did not acquire S-IgG, significant differences in BMI and eGFR were observed. This was equally true for the rate of S-IgG acquisition after the second vaccination that we previously reported [16]. Regarding BMI, low BMI was associated with failure of antibody acquisition in our transplant patient data. The median BMI of the Japanese post-transplant patients reported here was 22.9, which tended to be lower than that of Western post-transplant patients [28,29,30]. Previous reports have revealed that the acquisition rate of SARS-CoV-2 antibodies is associated with body weight, such as high BMI was associated with a low rate of antibody acquisition [24,31], and data have shown that the lower the BMI, the more susceptible to infection [32]. The association between antibody acquisition rate and BMI likely depends on both obesity and low BMI. Although BMI distribution differs among transplant recipients from Japan to that in the West, we speculated that the low body weight of Japanese post-renal-transplant patients might have adversely affected their acquisition of antibodies. Furthermore, considering the results of Table 3, there may be an optimal BMI, with a U-shaped association for antibody acquisition rate, rather than one that is linear. Regarding the relationship between eGFR and the antibody-acquisition rate, renal failure is known to reduce immunity [33]. Renal transplant patients had lower antibody levels than healthy participants and chronic kidney disease patients [34]. Therefore, particular attention should be focused on patients undergoing strong immunosuppressive therapy for rejection-related problems, among others, and those with impaired renal function. However, in our previous study [16], the longer the duration between transplantation and vaccination, the higher the antibody-acquisition rate tended to be. Conversely, in this study no correlation existed between the antibody-acquisition rate after the third vaccination and the duration between transplantation and vaccination. Immediately after transplantation, immunosuppression is strongest, and the immune response to the vaccine is potentially low [35]. However, this effect seems to diminish as time passes until the third vaccination.

In addition, S-IgG and NT-IgG were strongly correlated in Japanese post-renal transplant patients. It has been reported that when the neutralizing activity of S-IgG was measured in the general population infected with COVID-19, a certain number of NT-IgG-negative individuals were also found among the S-IgG-positive individuals [19]. Most of the S-IgG-positive renal transplant recipients in this study were NT-IgG-positive, except only two with low S-IgG antibody titers were NT-IgG-negative. These results suggest that Japanese renal transplant recipients are protective against COVID-19 infection if their S-IgG titer is elevated after vaccination.

As described above, both the antibody-acquisition rate and absolute antibody titer increased after a three-dose vaccination. Therefore, we recommend that Japanese kidney transplant recipients should receive at least three doses of the vaccine, as is the case overseas. Since the fourth vaccination may further increase the antibody-acquisition rate and absolute antibody titer, proceeding with additional vaccinations for post-kidney-transplant patients is plausible.

## 5. Limitations

This study had some limitations. First, the sample size was limited, and the study was conducted at a single center. Second, the timing of vaccination could not be controlled, and some participants could not identify the type of vaccine they received. Third, data were obtained exclusively from post-kidney transplant patients and were not compared with those of healthy controls. Fourth, we considered clinical characteristics only during the third antibody measurement; however, we could not account for their variation over time.

## 6. Conclusions

In Japanese post-kidney-transplant patients, three vaccine doses resulted in an increased antibody acquisition rate and absolute antibody titer. Furthermore, BMI and eGFR were associated with the rate of antibody acquisition after the third vaccination.

## Figures and Tables

**Figure 1 vaccines-11-00134-f001:**
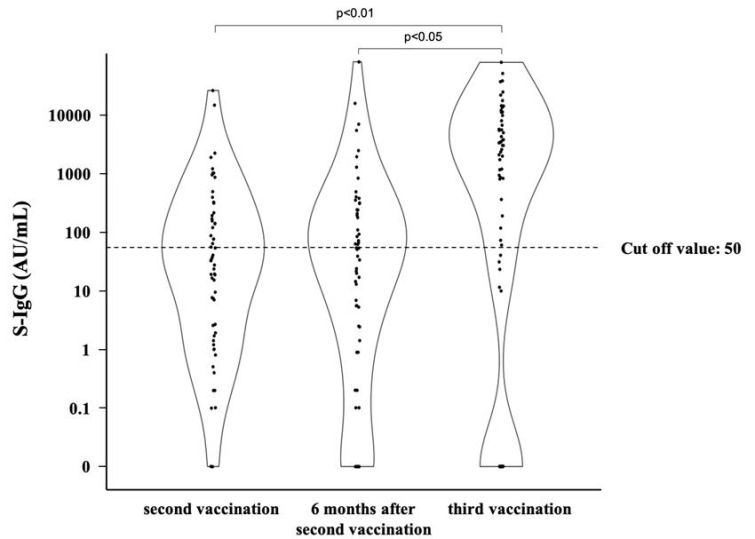
Changes in S-IgG titers in the post-kidney-transplant patients. The dots indicate the values of SARS-CoV-2 spike-IgG antibodies. The dotted line indicates the cutoff value.

**Figure 2 vaccines-11-00134-f002:**
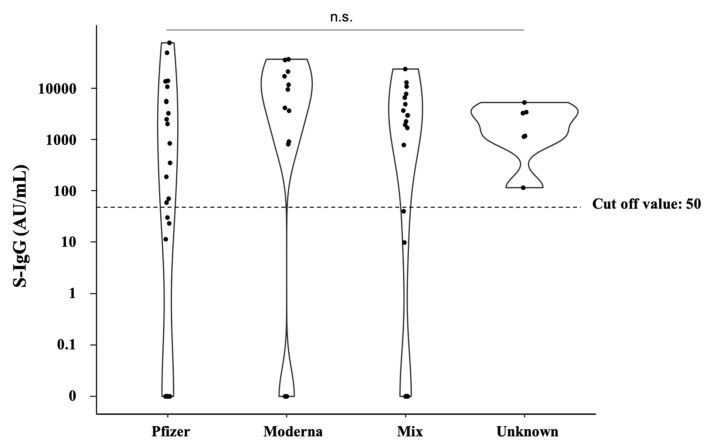
Comparison of S-IgG titers by vaccine type after the third vaccination. The dots indicate the values of SARS-CoV-2 spike-IgG antibodies. The dotted line indicates the cutoff value.

**Figure 3 vaccines-11-00134-f003:**
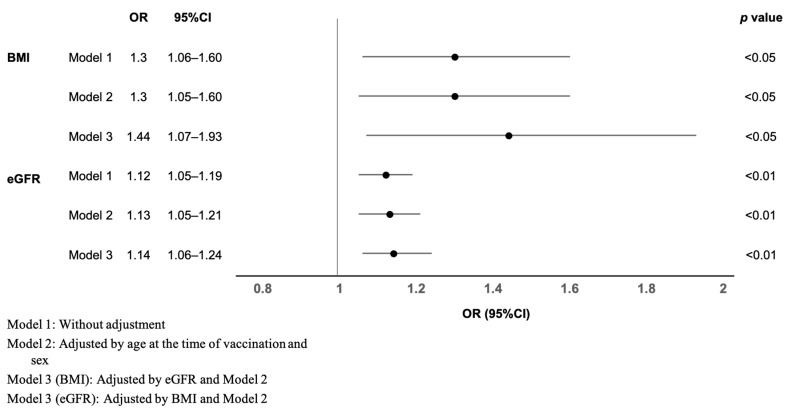
Clinical factors associated with S-IgG acquisition in post-kidney-transplant patients. OR: odds ratio; CI: confidence interval; BMI: body mass index; eGFR: estimated glomerular filtration rate.

**Figure 4 vaccines-11-00134-f004:**
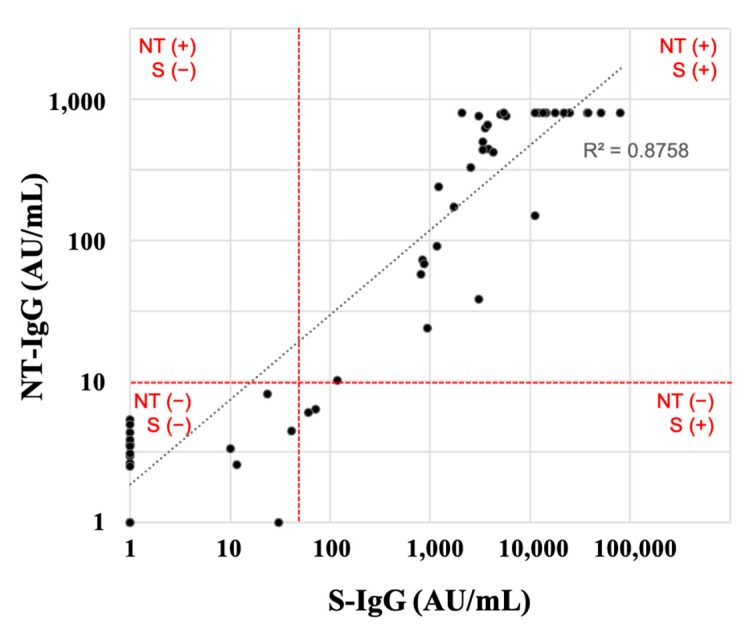
Correlation between NT-IgG and S-IgG after the third vaccination. The red line shows the cutoff value of NT-IgG and S-IgG, respectively. The dotted line shows the regression line. R, correlation coefficient; NT-IgG, neutralizing antibodies.

**Table 1 vaccines-11-00134-t001:** Clinical characteristics of study participants (*n* = 62).

Clinical Characteristics of Study Participants (*n* = 62)	
Age during vaccination, year, median [IQR]	54 [49–67]
Age during transplantation, year, median [IQR]	50 [38–58]
Female, *n* (%)	22 (35.5)
Height, cm, median [IQR]	163.5 [158.9–169.6]
Weight, kg, median [IQR]	64.6 [56.2–71.0]
BMI, kg m^2^, median [IQR]	22.9 [21.4–26.2]
eGFR, mL min^−1^ 1.73 m^2^, median [IQR]	42.8 [34.1–51.4]
The period between transplantation and vaccination, month, median [IQR]	84 [34–154]
Primary kidney diseases, *n* (%)	
Chronic glomerulonephritis	20 (32.3)
Diabetic nephropathy	9 (14.5)
Hereditary disease	8 (12.9)
Nephrosclerosis	4 (6.5)
Focal segmental glomerulosclerosis	1 (1.6)
Coexisting disease, *n* (%)	
Diabetes mellitus	15 (24.2)
Hypertension	39 (62.9)
Types of transplantation, *n* (%)	
Deceased donor kidney transplantation	6 (9.7)
Living-donor kidney transplantation	56 (90.3)
ABO-incompatible kidney transplantation	18 (29.0)
RRT before transplantation, *n* (%)	
Hemodialysis	33 (53.2)
Peritoneal dialysis	6 (9.7)
Preemptive kidney transplantation, *n* (%)	23 (37.1)
Pre-transplant RRT period, month, median [IQR]	17 [7–73]
Types of immunosuppressive drugs, *n* (%)	
Prednisolone	62 (100.0)
Mycophenolate mofetil	57 (91.9)
Tacrolimus	54 (87.1)
Cyclosporine	7 (11.3)
Types of SARS-CoV-2 vaccine, *n* (%)	
Pfizer-BioNTech	25 (40.3)
Moderna	12 (19.4)
Mix	19 (30.6)
Unknown	6 (9.7)

IQR, interquartile range; BMI, body mass index; eGFR, estimated glomerular filtration rate; RRT, renal replacement therapy; SARS-CoV-2, severe acute respiratory syndrome coronavirus 2.

**Table 2 vaccines-11-00134-t002:** Comparison between antibody-acquired and -nonacquired groups after the third vaccination.

	S-IgG (−) (*n* = 18)	S-IgG (+) (*n* = 44)	*p*-Value
Age during vaccination, year, median [IQR]	52 [49–67]	57 [50–68]	0.545
Age during transplantation, year, median [IQR]	47 [42–62]	51 [38–56]	0.704
Female, *n* (%)	6 (33.3)	16 (36.4)	1.000
Height, cm, median [IQR]	163.8 [158.1–169.6]	163.2 [159.5–169.1]	0.963
Weight, kg, median [IQR]	57.1 [52.0–65.3]	65.8 [58.4–73.4]	<0.05
BMI, kg m^2^, median [IQR]	21.3 [20.3–22.6]	24.2 [21.9–27.1]	<0.01
eGFR, mL min^−1^ 1.73 m^2^, median [IQR]	34.1 [25.9–40.6]	48.1 [37.6–54.5]	<0.01
The period between transplantation and vaccination, month, median [IQR]	77 [21–132]	90 [48–161]	0.285
Primary kidney diseases, *n* (%)			<0.05
Chronic glomerulonephritis	2 (11.1)	18 (40.9)	
Diabetic nephropathy	3 (16.7)	6 (13.6)	
Hereditary disease	4 (22.2)	4 (9.1)	
Nephrosclerosis	0 (0)	4 (9.1)	
Focal segmental glomerulosclerosis	0 (0)	1 (2.3)	
Coexisting disease, *n* (%)			
Diabetes mellitus	5 (27.8)	10 (22.7)	0.748
Hypertension	10 (55.6)	29 (65.9)	0.564
Types of transplantation, *n* (%)			
Deceased donor kidney transplantation	1 (5.6)	5 (11.4)	0.662
Living-donor kidney transplantation	16 (88.9)	40 (90.9)	1.000
ABO-incompatible kidney transplantation	5 (27.8)	15 (34.1)	1.000
RRT before transplantation, *n* (%)			0.710
Hemodialysis	9 (50.0)	24 (54.5)	
Peritoneal dialysis	1 (5.6)	5 (11.4)	
Preemptive kidney transplant, *n* (%)	8 (44.4)	15 (34.1)	
Pre-transplant RRT period, month, median [IQR]	23 [11–75]	16 [5–67]	0.298
Types of SARS-CoV-2 vaccine, *n* (%)			0.207
Pfizer-BioNTech	10 (55.6)	15 (34.1)	
Moderna	2 (11.1)	10 (22.7)	
Mix	6 (33.3)	13 (29.5)	
unknown	0 (0)	6 (13.6)	

IQR, interquartile range; BMI, body mass index; eGFR, estimated glomerular filtration rate; SARS-CoV-2, severe acute respiratory syndrome coronavirus 2.

**Table 3 vaccines-11-00134-t003:** Regression analysis for the increase in S-IgG titer from 5–6 months after 2 doses of vaccination to 3 doses of vaccination.

	*β*	95% CI	*T*	*p* Value
BMI Model 1	73.2	−888.1–1034.5	0.15	0.879
BMI Model 2	61.8	−920.1–1043.6	0.13	0.900
BMI Model 3	22.7	−943.2–988.7	0.05	0.963
eGFR Model 1	227.0	−49.5–503.7	1.64	0.106
eGFR Model 2	248.9	−37.5–535.2	1.74	0.087
eGFR Model 3	248.6	−40.9–538.0	1.72	0.090

BMI, body mass index; eGFR, estimated glomerular filtration rate; CI, confidence interval

## Data Availability

The datasets generated and analyzed during the current study are not publicly available because consent has not been obtained to make them available online to an unspecified number of people, but they are available from the corresponding author on reasonable request.

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
