# Peer review of "Assessment of Antibody-Titer Changes after Second and Third Severe Acute Respiratory Syndrome Coronavirus 2 mRNA Vaccination in Japanese Post-Kidney-Transplant Patients"

_vaccines, 2023, doi:10.3390/vaccines11010134_

Round 1
Reviewer 1 Report
The paper by Fujieda and colleagues is about the evaluation of the immune protection elicited by vaccines against COVID-19 in kidney transplant
recipients in Japan.
The introduction is very short, and could be written in a clearer form, as suggested in the minor points' section.
The methods section is too brief and provides only partially sufficient information.
The results section is also very brief, due to the single-technique nature of the study. It is not very clear, when association is presented between
antibody acquisition and clinical factors, if antibody quantitative measure has been used or only positivity/negativity. Please clarify.
More in general, since a semi quantitative technique has been used, please provide quantitative results along with percentage of positivity.
Another weak point of the results is that all comparison between groups are reported only for the last time point of the study.
Also the first or second time points could be equally investigated to better understand the differential behaviour (even if that was already done in the previous study,
because these are other subjects).
The discussion section is, in my humble opinion,of really low level on both the semantic and the biological point of view, and should be heavily modified and improved.
The limitations section is honest, but in my view most of these limitations could/should have been overcome before submitting the paper.
So, in my opinion, things to do to improve the paper and make it acceptable for publication could include:
- adding a second technique for antibody detection (e.g. neutralization assay)
- adding quantitative and kinetics data and using them in association studies)
- adding control subjects
- providing indication of time from vaccination as a further paramether to be assessed
- substantially improve the text in the evidenced areas (see next section)
Some minor points:
- line 27: e cutoff value is not a relevant element to be introduced in the abstract.
- line 38: Covid-19 pandemic was declared in early 2020, after the first autoctonous cases outside China were confirmed.
- lines 45-46: the sentence is unclear, please rephrase or choose a reference in support.
- lines 46-54: the period is written in unclear form, please rephrase. I guess the meaning is:' the effectiveness of vaccine protection
decays in time, so it is reccomendable the use of boosters, especially for immunocompromised subjects'.
- line 88: the paragraph should include some descritpion of the methods used for the analysis, not only the instrument and reagents.
- line 96: please explain the reason for this cut-off (reference to other studies, or to manufacturer's guidelines)
- line 141-142: please rephrase. It was not the antibody titers that were compared between groups, but other factors.
- line 144: primary disease is indicated as statistically significant, but in tab 2 there is not an exact indication of which of the clinical
conditions were significantly correlated to Ab positivity.
- line 144-146: fig 2 and its explanation is not correlated with the rest of this short paragraph and with its title
- line 147: please add the time point in the figure description
- line 170: a time frame ranging from 3 weeks to 3 months cannot be defined as 'immediately after'. Also, if we consider time from vaccination as a positive factor,
having no indication of this paramether for individual members of groups is a limitation (i.e.:are we sure that there was not a significant difference in time from vaccination
between antibody positive and negative subjects?) he same can be said about the difference between the results of different studies.
- line 172: please provide references to results of similar studies performed in other countries
- lines 172-180: the period is not supported by results obtained in the study, nor in the referred previous one, since in both studies no correlation has been
reported between age and antibody acquisition. So, despite the fact that in literature a correlation has been observed, in these studies it hasn't, and therefore it
can't be used for the interpretation of these studies' findings. Please modify the period accordingly.
- lines 186-193: the possibility that study subjects have been infected as asyntomatic carriers is real, and this is why, for example, an early measurement of antibodies should have
been performed (i.e. 1-2 weeks after the second or third dose), which could have identified the possbile concurrent infection.
Having no info about the possibility of each indivudual subject of having been infected is a limit to the study, please report it and discuss abou it.
- line 202: looks like the referred paper is about cancer, not infection. Please provide a more appropriate reference
- line 203: in both referred papers the association between bmi and antibody levels was indeed present, but with opposite trend with respect to this study. Please take
into consederation changing reference or rephrasing
- line 203-206: the period lacks any possible factual support and is merely speculative. Please rephrase
- line 210: the referred papers shows that ckd subjects have a much higher antibody level than transplant recipients. The subjects of your study are transplant recipients, even
if originally ckd, so please rephrase in a less equivocal form.
- line 217-233: This period compares results from this study and the previous one. Please, make use of the results of the new subjects for both time points and compare these.
- line 223-226: This period is badly written and biologically inexact. Please rephrase, including a more exact citation of the referred paper, or omit.
Reviewer 2 Report
In this study, the authors have analysed the effect on the antibody titers of a third vaccination in a series of 62 patients. They observed that a higher percentage of patients had a significant titer of antibody after the third injection.
- The authors have included a limited number of patients but they performed earlier a previous study including also a limited number of patients , to study the frequency of antibodies between the first and second injection. Why did they not continue to study this first cohort? It is highly probable that some patients of this first cohort have received a third dose.
- Fig2 : lines connecting dots should be better than only dots, so the reader could see the individual variations of the titer.
- Discussion, P line 186: the authors estimate that a possible asymptomatic infection could contribute to an increase of the antibody titer, but they can’t conclude because the patients had no monitoring by regular PCR. That is evident, but Abbott commercializes an assay which identify antibodies against the core (anti-N antibodies). The positivity of N-antibodies is an argument for an infection by SARS-CoV-2. Why the authors did not use this assay in patients with a very marked increase of anti-S antibodies?
- P 7, line 202: The authors suggest that the absence of a significant antibody levels is related to a low weight, and they explain (what is true) that Japanese people have a lower weight than “oversea” people. However, in the beginning of the discussion, line 172, they comment that the frequency of antibody is similar to that of other countries. Therefore, the argument concerning the low weight seems paradoxical and deserves further explanations.
Reviewer 3 Report
Evidently a follow up to the prior study of responses in a similar group of patients, this is an interesting but simplistic analysis of the antibody response to SARS-CoV-2 RBD S1 antigen in Japanese kidney transplant patients after receipt of a third dose of mRNA vaccine. Characterization of the patients is comprehensive, but measurement of their SARS-CoV-2 immune response is limited to data obtained using the Abbott semi-quantitative RBD S1 antibody binding assay. Interpretation of the utility of such antibodies in long term protection against SARS-CoV-2 variants is arguable and somewhat overstated in the introduction. The minimal data presented is, however, of interest in showing that a third dose of vaccine improves antibody production in this patient population. Specific suggestions:
1/ carefully check the wording in the introduction, there are a number of overstatements and grammatical errors. For example, "illustrating their usefulness in preventing infections and severe diseases". The data may suggest their usefulness in preventing infection and severe disease, but this has not been fully established. The contribution of T cell mechanisms remains to be clarified.
2/ Describe in more detail the assay to detect antibodies and its limitations (eg. does not detect a wide array of SARS-CoV-2 antibody specificities.
3/ Add statistics to the graphs presented.
4/ Proved a better explanation of the data presented in Figure 3 such as more explicitly how the models were derived. The x axis also needs a title.
Round 2
Reviewer 1 Report
I sincerely appreciate the authors' effort for improving their paper following my recommendations. I think they met the suggested requirements for acceptation, therefore I now propose to accept the paper for publication in the current form
Author Response
We thank you for your thoughtful suggestions and insights. The manuscript has benefited from your insightful remarks.